# Regulation of PHD2 by HIF-1α in Erythroid Cells: Insights into Erythropoiesis Under Hypoxia

**DOI:** 10.3390/ijms26020762

**Published:** 2025-01-17

**Authors:** Shunjuan Wang, Qiying Xu, Wenjing Liu, Na Zhang, Yuelin Qi, Feng Tang, Rili Ge

**Affiliations:** 1Research Center for High Altitude Medicine, Qinghai University, Xining 810016, China; 2Key Laboratory of the Ministry of High Altitude Medicine, Qinghai University, Xining 810016, China; 3Key Laboratory of Applied Fundamentals of High Altitude Medicine (Qinghai-Utah Joint Key Laboratory of Plateau Medicine), Qinghai University, Xining 810016, China; 4Laboratory for High Altitude Medicine of Qinghai Province, Qinghai University, Xining 810016, China

**Keywords:** erythroid differentiation, HIF-1α, endogenous PHD2 reporter systems

## Abstract

The hypoxia-inducible factor (HIF) pathway has been demonstrated to play a pivotal role in the process of high-altitude adaptation. PHD2, a key regulator of the HIF pathway, has been found to be associated with erythropoiesis. However, the relationship between changes in Phd2 abundance and erythroid differentiation under hypoxic conditions remains to be elucidated. A hemin-induced K562 erythroid differentiation model was used to explore the effects of PHD2 knockdown under hypoxia. Erythroid differentiation was assessed by flow cytometry and immunofluorescence. HIF-1α’s regulation of PHD2 was examined using luciferase assays and ChIP-seq. CRISPR/Cas9 was applied to knock out *EGLN1* and *HIF1A*, and a fluorescent reporter system was developed to track PHD2 expression. PHD2 knockdown enhanced erythroid differentiation, evident by increased CD71 and CD235a expression. Reporter assays and ChIP-seq identified an HIF-1α binding site in the *EGLN1* 5′ UTR, confirming HIF-1α as a regulator of PHD2 expression. The fluorescent reporter system provided real-time monitoring of endogenous PHD2 expression, showing that HIF-1α significantly modulates PHD2 levels under hypoxic conditions. PHD2 influences erythropoiesis under hypoxia, with HIF-1α regulating its expression. This feedback loop between HIF-1α and PHD2 sheds light on mechanisms driving erythroid differentiation under low-oxygen conditions.

## 1. Introduction

Hypoxia represents a significant challenge for humans living at high altitudes. Recent investigations into the acclimatization of lowlanders to hypoxia indicate that the hypoxia-inducible factor (HIF) pathway is crucial for adaptations including respiration, blood flow, vascular remodeling, and intermediary metabolism. HIF-1α has been identified as a critical factor in the development and progression of various diseases in numerous studies. HIF-1α contributes to the hypoxic microenvironment in solid tumors [1,2,3], is implicated in hypoxia-induced cardiovascular diseases [4,5], and plays a role in intermittent hypoxia’s protective effects on the cardiovascular and nervous systems [6], while also being involved in the inflammatory and immune responses associated with bacterial infections [7,8]. Key components of the HIF pathway, such as HIF-1α (hypoxia-inducible factor 1α), *EPAS1* (endothelial PAS domain-containing protein 1), *VHL* (von Hippel-Lindau protein), and *EGLN1* (egl-9 family hypoxia-inducible factor 1), have undergone positive selection in high-altitude-adapted populations, providing evidence of their critical involvement in human adaptation to hypoxia environments [9,10,11].

HIF-1α is maintained at low levels through post-translational modification but accumulates intracellularly under hypoxia conditions [12]. *EGLN1* encodes the PHD2 protein, a member of the proline hydroxylase family, which induces the degradation of HIF by hydroxylating Pro-402/564 in an oxygen-dependent manner [13,14]. PHD2 can modulate cellular metabolism extensively by hydroxylating key enzymes involved in metabolic and biosynthetic pathways [15,16]. Mutations in PHD2, particularly those leading to a partial loss of function, have been implicated in erythrocytosis, where excessive red blood cell production occurs, often in response to chronic hypoxia [17,18,19]. In investigating the mechanism by which PHD2 influences erythrocytosis, the enzyme activity of PHD2 was found to diminish in proportion to the reduction in oxygen content [20]. As discussed before, the impact of PHD2 mutations on erythroid progenitors remains contentious. Consequently, most studies on PHD2 have focused on altering its activity [21,22], yet the influence of PHD2 levels and its transcriptional control, especially in the context of erythroid progenitor differentiation under hypoxia conditions, remains underexplored [23,24]. To investigate these gaps, we used the CRISPR/Cas9 system to manipulate PHD2 expression and explore its role in regulating erythroid differentiation under normoxia and hypoxia conditions.

This study investigated the effects of *EGLN1* knockout on erythroid differentiation under hypoxia using a hemin-induced differentiation model. This study found that PHD2 abundance is modulated by HIF-1α in a time-dependent manner under hypoxia conditions. Additionally, we developed a reporter model using the CRISPR/Cas9 system to further elucidate the transcriptional regulation of PHD2 by HIF-1α, providing a detailed understanding of how HIF-1α controls PHD2 protein abundance in hypoxia environments. A HIF-1α knockout cell line further confirmed its role as the key regulator of PHD2 abundance in hypoxia environments.

## 2. Results

### 2.1. PHD2 Knockdown and Hypoxia Stimulation Promote Erythroid Differentiation

The effects of hypoxia stimulation and *EGLN1* on erythroid differentiation were examined using a hemin-induced erythroid differentiation model. Since CD71 and CD235a are cell-specific markers of erythroid precursors [25,26], their expression was analyzed using flow cytometry and immunofluorescence. Flow cytometry analysis of K562 cells after 72 h of hemin-induced differentiation showed a significant upregulation in the proportion of CD235a+ and CD71+ cells in PHD2 knockdown K562 cells under normoxia conditions (Figure 1a). Concurrently, the CD235a and CD71+ ratios in both K562 and K562-PHD2-ko cells showed a notable elevation under 1% PO_2_ hypoxia conditions. The impact of hypoxia stimulation and PHD2 knockdown on K562 cell differentiation showed no significant differences (Figure 1b). The mean fluorescence intensity (MFI) analysis demonstrated that both CD71 and CD235a expression levels were significantly enhanced following PHD2 knockdown in K562 cells. Additionally, hypoxia stimulation further augmented these expression levels, especially in the K562-PHD2-ko cells, where the upregulation of CD235a was more pronounced (Figure 1c). Immunofluorescence analysis corroborated these flow cytometry findings, confirming the increased expression of CD71 and CD235a in both PHD2 knockdown and hypoxia conditions (Figure 1d). Together, these results indicate that PHD2 plays a significant role in modulating erythroid differentiation, with its abundance influencing the differentiation of K562 cells under both normoxia and hypoxia conditions.

### 2.2. HIF-1α Regulate PHD2 Transcription via a Specific Motif

Given the significant effects of both PHD2 knockdown and hypoxia stimulation on erythroid differentiation, a hypoxia time-series model (0–48 h) was established to monitor changes in PHD2 abundance. Changes in HIF-1α and PHD2 protein levels under different hypoxia conditions were assessed by Western blot analysis. Western blot analysis revealed that HIF-1α protein levels increased significantly upon exposure to 1% oxygen, peaking at approximately 9 h before gradually declining. In parallel, PHD2 protein levels began to increase after 9 h of hypoxia exposure, peaking between 24 and 36 h before declining (Figure 2a,b). To determine whether the increased PHD2 protein abundance was due to elevated transcript levels, qPCR analysis was performed. The results showed that PHD2 mRNA levels increased approximately 2.5-fold at 6 h post-hypoxia exposure, stabilizing at roughly twice the normoxia levels thereafter (Figure 2c). HIF-1α ChIP-seq data from previous studies indicated that HIF-1α binds to the 5′ UTR of the *EGLN1* gene in certain cell types (e.g., PC-3 and T47D cells) [27,28,29], and H3K27ac data indicated that this region functions as an active enhancer, suggesting that HIF-1α may regulate *EGLN1* transcription by binding to this region (Figure 2d). To test this hypothesis, a dual-luciferase reporter assay was performed using a wild-type (WT) and mutant version of the *EGLN1* 5′ UTR region (Figure 2e). HIF-1α overexpression led to a significant increase in luciferase activity in both normoxia and hypoxia conditions. However, mutation of the HIF-1α binding motif within the 5′ UTR region resulted in a marked reduction in luciferase activity, confirming the importance of this motif for HIF-1α-mediated regulation of PHD2 transcription (Figure 2f).

### 2.3. Establishment of a Fluorescent Reporter Model for Endogenous PHD2 Expression

To enable real-time monitoring of endogenous PHD2 expression, a fluorescent reporter system was developed by integrating a T2A-eGFP gene into the 3′ UTR of the *EGLN1* gene using CRISPR/Cas9 gene editing (Figure 3a). Following cell transfection, fluorescence-positive cells were sorted by FACS and clonally expanded to generate monoclonal cell lines. Flow cytometric analysis of these monoclonal cell lines revealed a distinct green fluorescence peak in cells carrying the PHD2-T2A-eGFP knock-in allele (Figure 3c). PCR and Sanger sequencing confirmed the correct insertion of eGFP into the target site (Figure 3d,e). Confocal microscopy further validated the fluorescence expression in the cytoplasm of the KEeC3 cell line (Figure 3f). This reporter system provides a reliable method for monitoring PHD2 expression in real time.

### 2.4. HIF-1α Regulates Endogenous PHD2 Expression Under Normoxia Conditions

To investigate the impact of HIF-1α on endogenous PHD2 expression, the KEeC3 cell line was subjected to hypoxia conditions (1% O_2_) over a time course, as previously established (Figure 4a). Flow cytometry analysis of FITC fluorescence intensity demonstrated a strong correlation between PHD2 expression levels and eGFP fluorescence intensity in response to hypoxia (Figure 4b). To investigate the effect of HIF-1α on endogenous PHD2 protein abundance, a KEeC3 cell line with HIF-1α knockdown (KEeC3-1αKO) was generated. Moreover, when HIF-1α was knocked down in KEeC3 cells, a significant reduction in fluorescence intensity was observed, both by flow cytometry and confocal microscopy, indicating that HIF-1α plays a key role in regulating PHD2 expression in these cells (Figure 4c–e).

### 2.5. HIF-1α Is a Major Transcriptional Regulator of PHD2 in Hypoxia

Finally, to examine whether HIF-1α is a major factor regulating the expression of PHD2 under hypoxia conditions, a HIF-1α knockout (KO) cell line was generated in K562 cells (Figure 5a). Consistent with previous experimental results, the cells were cultured in a 1% O_2_ environment for 7 h (the peak of HIF-1α expression) or 20 h (the peak of PHD2 expression). Western blot analysis showed that PHD2 protein levels in the three HIF-1α knockout cell lines did not change significantly before or after hypoxia stimulation, unlike the K562 WT cells, where PHD2 protein levels increased markedly under hypoxia stimulation (Figure 5b,c). Furthermore, RT-qPCR results indicated that the transcription of PHD2 was no longer responsive to hypoxia in HIF-1α KO cells, suggesting that HIF-1α is essential for the transcriptional regulation of PHD2 in hypoxia (Figure 5d).

In the hemin induction model, HIF-1α KO did not significantly alter the erythroid differentiation potential of K562 cells. However, PHD2 KO and HIF-1α knockdown induced a higher differentiation rate, suggesting that these pathways act synergistically to regulate erythropoiesis under hypoxia conditions (Figure 5e,f). Collectively, these results confirm that HIF-1α is a crucial regulator of PHD2 expression and that both HIF-1α and PHD2 contribute to the regulation of erythropoiesis under hypoxia stress. These findings suggest that the increase in PHD2 abundance compensates for the accumulation of HIF-1α resulting from reduced enzyme activity, thereby completing the negative feedback loop from HIF-1α to PHD2 (Figure 5g).

## 3. Discussion

This study demonstrates that PHD2 knockdown stimulates erythropoiesis as effectively as hypoxia, using a hemin-induced K562 erythropoiesis differentiation model. We demonstrate that PHD2 expression is regulated by HIF-1α in a time-dependent manner under hypoxia conditions. This regulation occurs via direct binding of HIF-1α to the 5′ UTR region of the *EGLN1* gene, leading to increased transcription of PHD2 and its subsequent accumulation.

In recent decades, the cellular hypoxia-inducible factor (HIF) pathway has been implicated in the adaptation of mammals to high-altitude hypoxia environments [30,31]. A consistent outcome of acclimatization and adaptation to hypoxia is an increase in hematocrit (Hct) and hemoglobin concentration, which facilitate oxygen transport in the body [32]. In a previous study, our group found that Tibetan populations carrying the D4E:C127S mutation in the PHD2 gene exhibited lower hemoglobin levels at high altitudes [33,34]. Tibetan Phd2 mice, engineered by Frank S. Lee’s team, exhibit an enhanced hypoxia ventilatory response and normal levels of hematocrit and hemoglobin [18,22,35]. Conversely, a recent characterization of genetic variants in the *EGLN1* gene, identified in a cohort of European erythrocytosis patients, revealed a link between this protein and erythrocytosis [19]. Similarly, we observed that PHD2 knockdown significantly induced erythrocyte differentiation (Figure 1). Figure 1a shows that the proportion of CD71⁺ CD235a⁺ double-positive cells is similar after PHD2 knockdown and hypoxic stimulation (85.9% vs. 84.7%), but the mean fluorescence intensity differs in Figure 1c. This discrepancy may be due to the activation of additional erythrocyte differentiation factors during hypoxia, altered HIF-1α expression, and the involvement of other regulators like HIF-2α and HIF-3α. Figure 1c has been updated for clearer data presentation.

Previous studies on PHD2 have focused on its hydroxylase activity, utilizing oxygen as a substrate [36]. Few studies, however, have explored its transcriptional regulation. PHD2 abundance has been shown to be regulated at both the translational and transcriptional levels by Death-associated protein 5 (DAP5) in HeLa cells [23] and estrogen receptor beta (ESR2) in epithelial cells [24]. However, both *EGLN1* and *EGLN3* are induced by hypoxia at the transcriptional level in a HIF-dependent manner [37]. Gisela D’Angelo et al. demonstrated that transcriptional blockade with actinomycin D resulted in enhanced PHD2 enzyme activity under in vitro hypoxia conditions [36]. In the present study, a similar pattern was observed in the hypoxia time-series model established using human chronic myeloid leukemia K562 cells. The abundance of HIF-1α initially increased, peaked, and then decreased over time, concomitant with an increase in PHD2 levels (Figure 2a). This increase has been confirmed to be attributed to the elevated level of the transcription factor (Figure 2c). A subsequent review of previous HIF-1α ChIP-seq databases [27,28,29] revealed that HIF-1α binds to the *EGLN1* 5′UTR in PC-3 and T47D cells, producing a distinct peak (Figure 2d). A dual-luciferase reporter assay was designed based on this binding site to confirm whether HIF-1α initiates transcription of the *EGLN1* gene in K562 cells by binding to it, and the results confirmed this interaction (Figure 2e,f).

Previous research has shown that, although the *EGLN1* UTR contains a binding site for HIF-1α, HIF-1α’s regulation of *EGLN1* transcription is not consistent across cells from different tissues [38,39,40]. Additionally, HIF-1α ChIP-seq data showed that HIF-1α does not always bind to the 5′ UTR of *EGLN1* [29]. This phenomenon may be attributed to epigenetic modifications present in cells from different tissues and under varying environmental conditions. To enable real-time monitoring of intracellular PHD2 abundance, an endogenous PHD2 expression fluorescence reporter model (KEeC3) was established in this study. The T2A sequence, which can be cleaved after translation [41], was inserted between PHD2 and the eGFP fluorescent proteins to prevent fusion expression from affecting PHD2 protein activity (Figure 3). The results from FAM assay and Western blot analysis in the KEeC3 model demonstrated that the system could effectively respond to PHD2 protein abundance, as evidenced by changes in eGFP fluorescence intensity (Figure 4a,b). After the knockdown of HIF-1α in KEeC3 cells, a significant reduction in mean fluorescence intensity (MFI) was observed by both flow cytometry and confocal microscopy. These results clearly demonstrated that HIF-1α promotes *EGLN1* transcription, thereby increasing PHD2 abundance (Figure 4c–e).

The most extensively characterized HIF isozymes, HIF-1α and HIF-2α, are known to stimulate the transcription of hundreds of genes essential for the adaptive responses to hypoxia [42]. Despite their significant similarities, the roles of these two isozymes do not entirely overlap. HIF-1α knockdown cell lines were constructed and cultured using the CRISPR system, as shown in Figure 2a–c. These cells were exposed to 1% oxygen for either 7 or 20 h. The transcriptional and protein levels of PHD2 remained unchanged in HIF-1α knockdown cells following hypoxia stimulation, suggesting that HIF-1α is the primary regulator of PHD2 abundance under hypoxia conditions.

The PHD2 molecule is a key component of the hypoxia signaling pathway, serving as an oxygen concentration receptor. Its protein abundance varies with changes in oxygen levels and the duration of hypoxic exposure, a process likely regulated by multiple factors over time. In this study, we focused on the transcriptional regulation of PHD2 by HIF-1α during acute hypoxic stimulation, but the post-transcriptional and post-translational mechanisms remain unexplored. The reporter model developed in this study, which tracks endogenous PHD2 protein abundance, provides a solid foundation for identifying additional factors that regulate PHD2 levels. In future work, we plan to investigate other potential regulatory factors using established molecular biology techniques, informed by similar studies and complemented by high-throughput sequencing.

## 4. Materials and Methods

### 4.1. Plasmids and Cloning

PX459-pSpCas9(BB)-2A-Puro-V2.0 (Addgene Watertown, MA, USA) [43] was used for sgRNA and Cas9 delivery. The sgRNAs for *EGLN1* exon5 were designed by online tools (URL accessed on 10 August 2022 at: http://crispor.tefor.net/crispor.py and URL accessed on 12 August 2022 at: http://chopchop.cbu.uib.no/) according to the genomic sequence NG_015865.1 and synthesized by Sangon Biotech (Sangon Biotech, Shanghai, China). The homologous recombinant donor for targeted knock-in was cloned into pUC19 (ThermoFisher, Waltham, MA, USA). The T2A-eGFP and PGK-HSV-TK sequences were synthesized by Sangon Biotech, and the two homology arms were amplified from the genome by PCR. The sgRNA sequences used in this study were as follows: *EGLN1* exon5-KI-sgRNA1 for 5′-caccgactcaataaaccttcagatt-3′, rev 5′-aaacaatctgaaggtttattgagtc-3′; *EGLN1* exon5-KI-sgRNA2 for 5′-caccgaataaaccttcagattcggt-3′, rev 5′-aaacaccgaatctgaaggtttattc-3′; *EGLN1* KO-sgRNA1 for 5′-caccggccgcgtcgccgtgtcgtg-3′, rev 5′-aaaccacgacacggcgacgcggcc-3′; *EGLN1* KO-sgRNA2 for 5′-caccgcgcgccgggacaacgcctc-3′, rev5′-aaacgaggcgttgtcccggcgcgc-3′; *HIF1A* KO-sgRNA1 for 5′-caccgcataatgtgagttcgcatct-3′, rev5′-agatgcgaactcacattatgcggtg-3′; *HIF1A* KO-sgRNA2 for 5′-caccgttgataaggcctctgtgatg-3′, rev5′-catcacagaggccttatcaacggtg-3′.

### 4.2. Mammalian Cell Culture

The K562 (ATCC, CCL-243) cell line was verified by a series of SNP analyses and subsequently determined to be mycoplasma-free by routine tests. The cells were maintained in a humidified incubator at 37 °C in an atmosphere of 5% CO_2_ with the addition of Roswell Park Memorial Institute (RPMI) 1640 culture medium (Gibco, Waltham, MA, USA)supplemented with 10% FBS (HyClone, Marlborough, MA, USA), 1% L-glutamine (Gibco, Waltham, MA, USA) and 1% penicillin/streptomycin (Gibco, Waltham, MA, USA). The cell lines within 10 passages of their initial thawing were used for experiments.

The generation of K562 cells with PHD2-T2A-eGFP was achieved through the electroporation of up-HA-T2A-eGFP-down-HA-PGK-HygR and pX459-*EGLN1* sgRNA into human lymphoblastoid K562 cells, followed by the isolation of GFP-positive cells through fluorescence-activated cell sorting (FACS). The K562 eGFP-positive cells were subjected to screening with the antiviral drug ganciclovir (C_6_H_13_NO_5_) (ThermoFisher, Waltham, MA, USA). Individual clones were generated and amplified using limited dilution. The T2A-GFP knock-in was verified by genomic PCR and FAM. K562 with PHD2-T2A-eGFP clone E was selected for screening and validation.

The generation of K562 cells with PHD2 or HIF1α knockout was achieved through the electroporation of u pX459-*EGLN1/HIF1A* sgRNA into human lymphoblastoid K562 cells, followed by screening with the antibiotic puromycin (MCE, Monmouth Junction, NJ, USA). Individual clones were generated and amplified using limited dilution. PHD2 or HIF1α knockout was verified by immunoblot.

### 4.3. Fluorescence-Activated Cell Sorting (FACS)

Flow fluorescence analysis and cell sorting were conducted using BD FACS Aria™ III (BD Biosciences, Franklin Lakes, NJ, USA). Cell debris was excluded based on forward scatter (FSC-A) and side scatter (SSC-A) parameters. Single cells were identified using FSC-A and forward scatter height (FSC-H) measurements. To distinguish live cells, those negative for AF700 (Becton, Dickinson and Company, Franklin Lakes, NJ, USA) were selected. Conversely, cells positive for FITC (488 nm/509 nm) were identified as eGFP-expressing cells. The positive cells were collected using tubes containing 10% fetal bovine serum (HyClone, Marlborough, MA, USA), penicillin (100 U/mL) and streptomycin (100 μg/mL) RPMI-1640 (Gibco, Waltham, MA, USA). PE mouse anti-human CD71 (M-A712), APC mouse anti-human CD235α (GA-R2(HIR2)), PE mouse IgG2α, κ isotype control (G155-178), PE mouse IgG2α, κ isotype control (G155-378), and APC mouse IgG2α, κ isotype control (27–35) were purchased from BD Biosciences (BD Biosciences, Franklin Lakes, NJ, USA). Data analysis was conducted using FlowJo10 software.

### 4.4. Immunoblot and Antibodies

Total protein was obtained by lysing the cells with RIPA buffer (Sigma-Aldrich, St. Louis and Burlington, MA, USA, Life Science, Tokyo, Japan) and protease inhibitors, which were used during the Western blotting process. The concentration of the extracts was determined using the Pierce™ BCA Protein Assay Kit (ThermoFisher, Waltham, MA, USA). The primary antibodies used were HIF-1α (Abcam, Cambridge, UK), PHD2 (CST, Boston, MA, USA), β-Actin (Servicebio, Wuhan, China). The secondary antibodies used were polyclonal goat anti-mouse IgG H&L HRP (Abcam, Cambridge, UK) and polyclonal goat anti-rabbit IgG H&L HRP (Abcam, Cambridge, UK). The strip buffer used was Restore Western Blot Stripping Buffer (ThermoFisher, Waltham, MA, USA).

### 4.5. Quantitative Real-Time Polymerase Chain Reaction (qRT-PCR)

Total RNA was isolated from preosteoclasts using a total RNA extraction kit (TIANGEN Biotech, Beijing, China) and subjected to cDNA synthesis using FastKing gDNA Dispelling RT SuperMix (TIANGEN Biotech, Beijing, China). Quantitative real-time polymerase chain reaction (qRT-PCR) was conducted on a QuantStudio 5 flex (ThermoFisher, Waltham, MA, USA) using QuantiNova SYBR Green PCR kit (Qiagen, Hilden, Germany). Relative quantification calculations were performed using the 2^−ΔCT^ method. The gene-specific primer sets were as follows: *HIF1A*, for 5′-ccacaggacagtacaggatg-3′, rev 5′-tcaagtcgtgctgaataatacc-3′; *EGLN1*, for 5′-tgcgaaaccattgggctgct-3′, rev 5′-gtcacacatcttccatctcc-3′; β-Tubulin, for 5′-ctctgaagctgaccacacca-3′, rev 5′-gccaggcataaagaaatgga-3′.

### 4.6. Dual-Luciferase Reporter Assay

The regulatory activity of HIF-1α on the EGLN1 gene was quantified by a dual-luciferase reporter assay. K562 cells were co-transfected with pGL4.10 [luc2] (Promega, Madison, WI, USA), carrying the EGLN1 5′UTR sequence of HRE/HRE-mut, and pGL4.74 (Promega, Madison, WI, USA). Following a 24 h incubation period, half of the cells were transferred to a 1% oxygen incubator for a further 7 h. Thereafter, fluorescence intensity was measured in conjunction with the cells cultured in normoxia, in accordance with the instructions set out in the Dual-Luciferase^®^ Reporter Assay System technical manual (Promega, Madison, WI, USA). The intensity of the HIF-1α gene was estimated and correlated by calculating the ratio of the bioluminescent signals of firefly luciferase (Fluc) and Renilla luciferase (Rluc).

### 4.7. Statistical Methods

All data were analyzed using GraphPad Prism 9.0 (GraphPad Software) and presented as “mean ± standard deviation (SD)”. Comparisons between two groups were performed using the *t*-test, while multiple comparisons between groups were analyzed using Two-way ANOVA. Statistical significance was defined as *p* < 0.0.

## 5. Conclusions

In conclusion, this study confirms that PHD2 knockout is as effective as hypoxia in stimulating erythropoiesis. An endogenous PHD2 expression reporter model was established, which will aid in future studies on regulators of PHD2 protein abundance. Furthermore, our findings support the hypothesis that HIF-1α plays a pivotal role in regulating PHD2 levels in hypoxia environments.

## Figures and Tables

**Figure 1 ijms-26-00762-f001:**
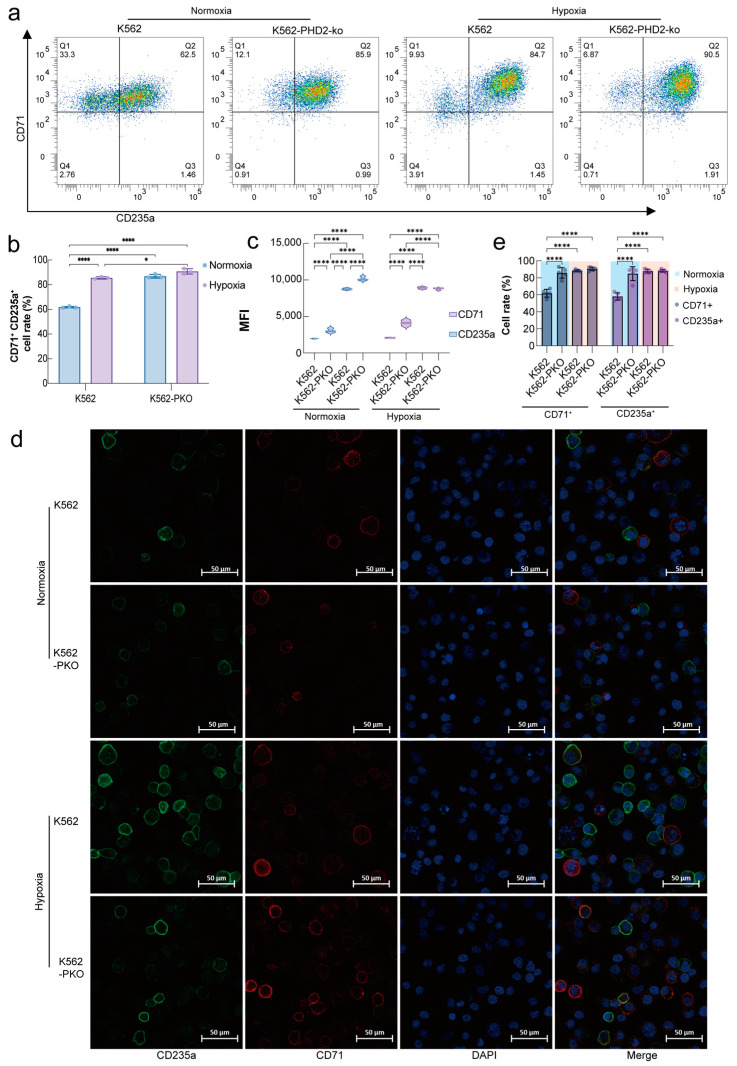
PHD2 knockout and hypoxia stimulation promote erythroid differentiation. (**a**) Flow assay analysis for CD71 and CD235a expression changes after hemin induction. The cells were analyzed after being under normoxia or hypoxia (1% O_2_) conditions for 72 h. (**b**) The quantitative results of flow cytometry analysis are presented as a rate of positive cells expressing CD71 and CD235a. (**c**) The quantitative results of the flow cytometry analysis are expressed as mean fluorescence intensity for both PE-CD235a and APC-CD71. (**d**) Confocal laser analysis of CD71 and CD235a expression after hemin induction for 72 h. The data are demonstrated as mean ± standard deviation (S.D.) and represent three independent experiments. (**e**) The quantitative results of confocal laser analysis are presented as a rate of positive cells expressing CD71 or CD235a. Statistical analyses were accomplished using a one-way ANOVA comparison test. * *p* < 0.05; **** *p* < 0.0001.

**Figure 2 ijms-26-00762-f002:**
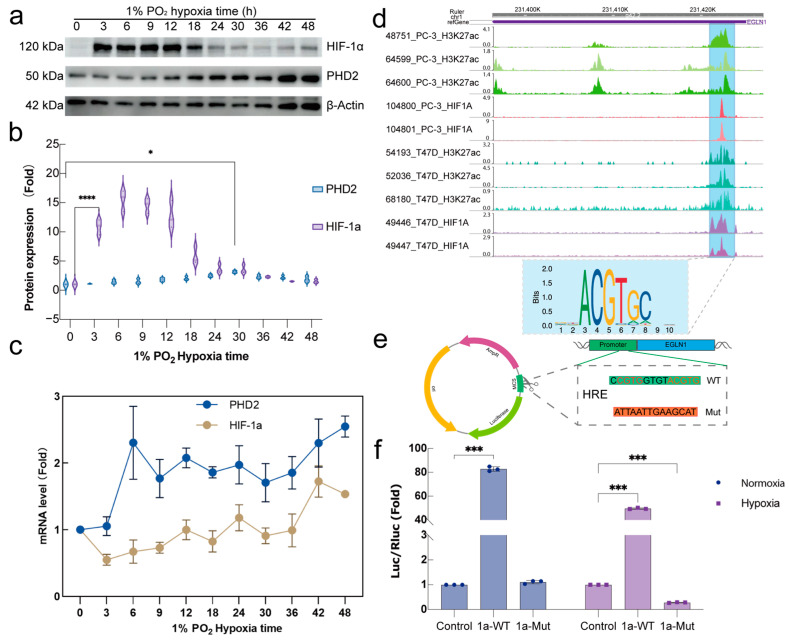
HIF-1α regulates PHD2 transcription through 5′ UTR HRE. (**a**–**c**) Protein expression (**a**) and mRNA level (**c**) of PHD2 and HIF-1α. K562 cells were stimulated by 1% O_2_ for 0–48 h. To determine the expression level of PHD2 and HIF-1α, cells were harvested for immunoblotting (**a**,**b**) and qPCR (**c**). (**d**) HIF1α and H3K27 Chip-seq data analysis of HIF1α binding site and H3K27 acetylation site confirms EGLN1promoter region. (**e**) Schematic diagram of the constructed dual-luciferase reporter plasmid. WT: wild type; Mut: HRE site mutant. (**f**) Determination of transcriptional regulatory activity by dual-luciferase reporter assay. The data are demonstrated as mean ± standard deviation (S.D.) and represent three independent experiments. Statistical analyses were accomplished using a one-way ANOVA comparison test. * *p* < 0.05; *** *p* < 0.001; **** *p* < 0.0001.

**Figure 3 ijms-26-00762-f003:**
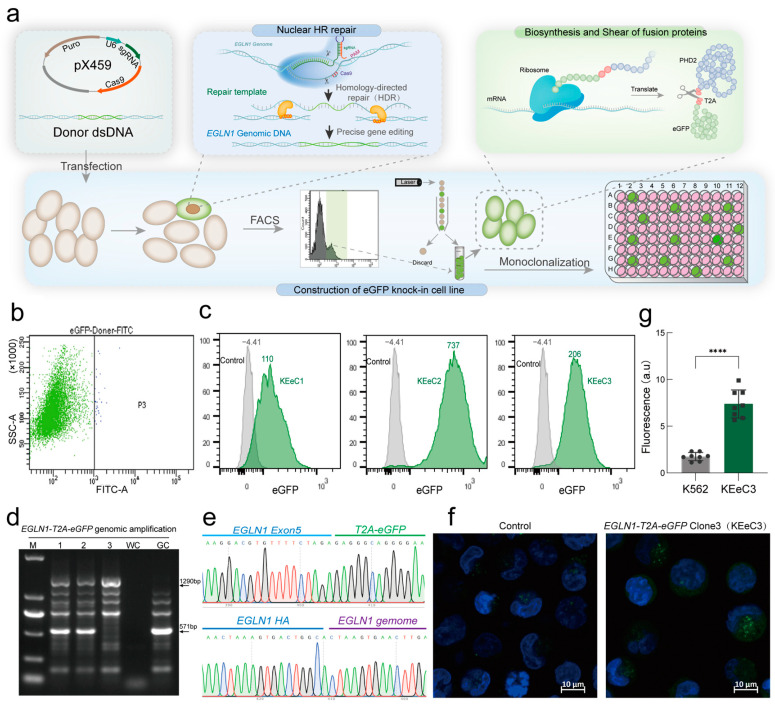
Establishment of a fluorescent reporter model for endogenous PHD2 expression. (**a**) Establishment of the fluorescent reporter model for endogenous PHD2. K562 cells were transduced with pX459-*EGLN1* sgRNA and *EGLN1*-HA-eGFP donor vector. After puromycin selection, FITC positive cells were sorted by FACS. The monoclonalization of sorted cells was achieved using the finite dilution method. (**b**) K562 cells were subjected to flow cytometry analysis to sort those exhibiting a fluorescence intensity exceeding 10^3^. (**c**) Confirmation of clonized FITC-positive cells by flow cytometry. (**d**) PCR amplification to verify whether the resulting FITC-positive clone was an *hEGLN1* exon5 eGFP Knock-in holomorphic, 1~3, *EGLN1*-eGFP Clone1~3 genome as a template, GC, K562 cell genome as a template, and WC, negative control of untemplated PCR. (**e**) Results of Sanger sequencing products versus the *hEGLN1* genome sequence matching results. (**f**) Detection of fluorescent protein expression in eGFP knock-in cells by laser confocal microscopy. (**g**) The quantitative results of confocal laser analysis are presented as mean fluorescence intensity. Statistical analyses were accomplished using *T*-test. **** *p* < 0.0001.

**Figure 4 ijms-26-00762-f004:**
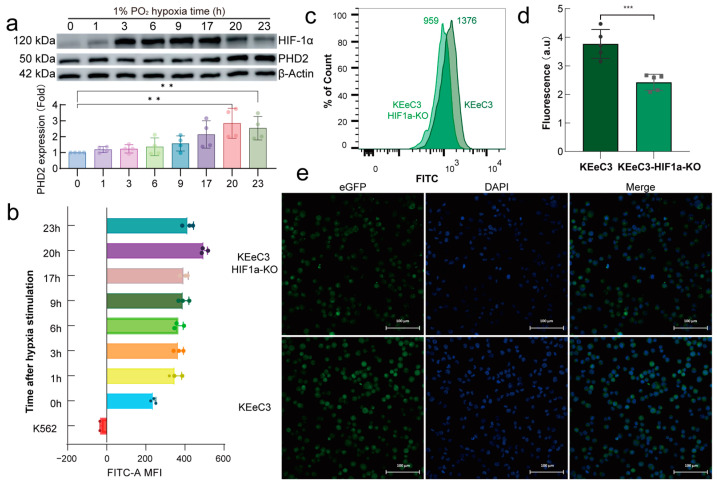
HIF-1α regulates PHD2 expression in K562 cells under normoxia. (**a**) Protein expression of PHD2 and HIF-1α. KEeC3 cells were stimulated by 1% O_2_ for 0–23 h. To ascertain the levels of protein expression and fluorescence intensity, the cells were harvested and subjected to immunoblotting (**a**) and flow cytometry (**b**), respectively. (**c**) Flow cytometry analysis of changes in fluorescence intensity of the HIF-1α KO KEeC3 cell population. (**d**,**e**) Images and quantitative results of confocal cell mean fluorescence intensity. The data are demonstrated as mean ± standard deviation (S.D.) and represent three independent experiments. Statistical analyses were accomplished using a one-way ANOVA comparison test. ** *p* < 0.01; *** *p* < 0.001.

**Figure 5 ijms-26-00762-f005:**
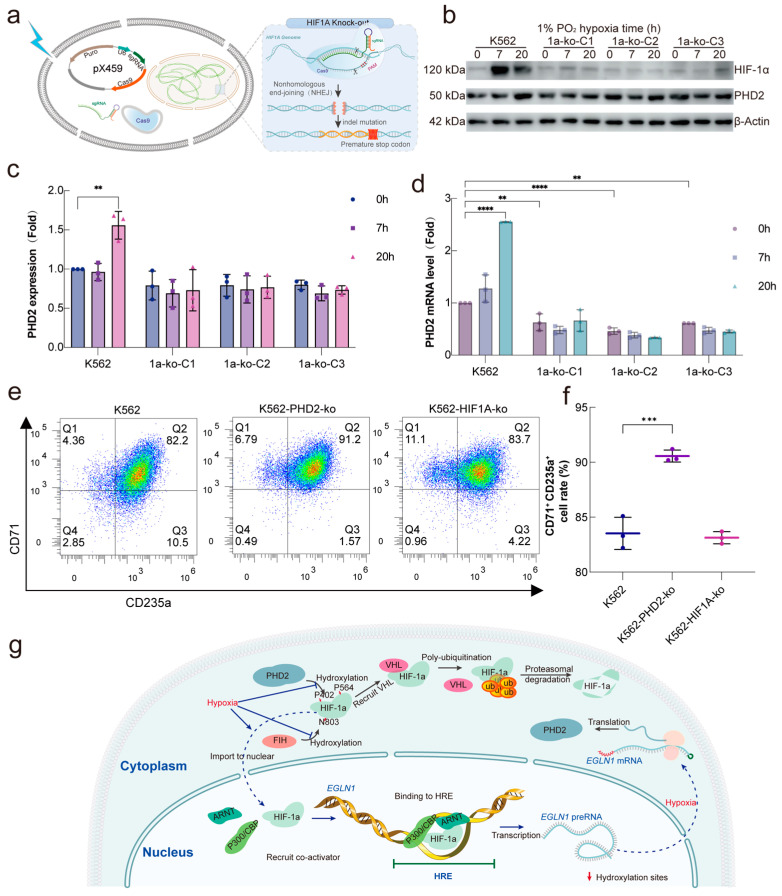
HIF-1α is a major transcriptional regulator of PHD2 under hypoxia conditions. (**a**) Schematic diagram of the constructed HIF-1α knockout K562 cell line. (**b**–**d**) Protein expression of PHD2 and HIF-1α of HIF-1α knockout K562 cell line was stimulated by 1% O_2_ for 7 h or 20 h; cells were harvested for immunoblotting (**b**,**c**) and qPCR (**d**). (**e**) An assay examined CD71 and CD235a expression following hemin induction for 72 h. Cells were analyzed under both normoxia and hypoxia conditions. (**f**) Flow cytometry results were expressed as the percentage of CD71- or CD235a-positive cells. The data are presented as mean ± standard deviation (S.D.) and represent three independent experiments. Statistical analyses were accomplished using a one-way ANOVA comparison test. ** *p* < 0.01; *** *p* < 0.001; **** *p* < 0.0001. (**g**) The model diagram illustrates the regulation of PHD2 abundance in diverse oxygen partial-pressure environments.

## Data Availability

Data are contained within the article.

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
