# Peer review of "Regulation of PHD2 by HIF-1α in Erythroid Cells: Insights into Erythropoiesis Under Hypoxia"

_ijms, 2025, doi:10.3390/ijms26020762_

Round 1
Reviewer 1 Report
Comments and Suggestions for Authors
The hypoxia-inducible factor (HIF) pathway plays a significant role in adapting at high altitudes via erythropoiesis. HIF and PHD2 are playing key role in erythroid differentiation. CRISPR/Cas9, ChIP-seq, and fluorescent reporters are used in the study to show how HIF-1α regulates the expression of PHD2. This helps us understand how erythroid differentiation works when oxygen levels are low and how it is affected by PHD2.
The study is thoughtfully designed, and the experiments are meticulously conducted, effectively addressing the primary objective of the manuscript. Authors need to address the following comments.
1. For Figure 1c, the author should clarify the rationale behind the observation: if PHD2 is an essential regulator of HIF-α, then PHD2 knockdown should theoretically enhance erythropoiesis to a similar extent as hypoxic conditions. This connection needs to be explicitly addressed.
2. Figure 1d, 3f and 4e is missing quantitative data to support the findings. As the experiments were performed three times, it is essential to include individual values to validate the significance and reproducibility of the representative image shown.
3. The molecular weight unit on the blots is incorrect, it is written as "kd" but should be "kDa." The internal controls used in the blots are β-actin (42 kDa) and α-tubulin (55 kDa), both of which are very close to the PHD2 band (50 kDa). Were the internal control blots run separately, or were the membranes stripped and re-probed after detecting the target protein? Using two different internal controls (β-actin and α-tubulin) is inconsistent. Unless there is a specific justification, it is better to use only one kind of internal control. Moreover, a more suitable internal control in this case would be GAPDH (36 kDa), which is sufficiently distant from the PHD2 band.
4. In Figures 1b and 5F, if the author is representing the Q2 population, the use of a slash ("/") between the two markers is inappropriate. The y-axis should be labeled as CD71⁺CD235a⁺, not CD71⁺/CD235a⁺, to correctly represent the double-positive population.
Author Response
Comments 1: For Figure 1c, the author should clarify the rationale behind the observation: if PHD2 is an essential regulator of HIF-α, then PHD2 knockdown should theoretically enhance erythropoiesis to a similar extent as hypoxic conditions. This connection needs to be explicitly addressed.
Response 1: Thank you for your suggestion and we are glad to receive your comments. We totally agree with you. From Figure 1a, we observe that the proportion of CD71⁺ CD235a⁺ double-positive cells after PHD2 knockdown is similar to that after direct hypoxic stimulation (85.9% versus 84.7%). However, a difference is evident in the mean fluorescence intensity shown in Figure 1c. The discrepancy observed may be explained by several factors. First, during hypoxic stimulation, the entire hypoxic signaling pathway is activated, including erythrocyte differentiation factors that are not regulated by PHD2, which could account for the difference. Second, compared to PHD2 knockdown under normoxia, HIF-1α expression was altered after hypoxic exposure. Finally, while HIF-1α is the primary target of PHD2, other regulators, such as HIF-2α and HIF-3α, are also involved in the hypoxic response and may be abnormally expressed under hypoxic conditions. These molecules are also involved in erythrocyte cell differentiation. Additionally, we have updated Figure 1c for clearer presentation of the data. This explanation has been added to lines 240-245 of the manuscript.
“Figure 1a shows that the proportion of CD71⁺ CD235a⁺ double-positive cells is similar after PHD2 knockdown and hypoxic stimulation (85.9% vs. 84.7%), but the mean fluorescence intensity differs in Figure 1c. This discrepancy may be due to the activation of additional erythrocyte differentiation factors during hypoxia, altered HIF-1α expression, and the involvement of other regulators like HIF-2α and HIF-3α. Figure 1c has been updated for clearer data presentation.”
Comments 2: Figure 1d, 3f and 4e is missing quantitative data to support the findings. As the experiments were performed three times, it is essential to include individual values to validate the significance and reproducibility of the representative image shown.
Response 2: We also want to thank you for your constructive suggestions to enhance the rigor of our manuscript. In response, we have included additional quantitative data in the revised manuscript, specifically from Figures 1e, 3g, and 4d.
Comments 3: The molecular weight unit on the blots is incorrect, it is written as "kd" but should be "kDa." The internal controls used in the blots are β-actin (42 kDa) and α-tubulin (55 kDa), both of which are very close to the PHD2 band (50 kDa). Were the internal control blots run separately, or were the membranes stripped and re-probed after detecting the target protein? Using two different internal controls (β-actin and α-tubulin) is inconsistent. Unless there is a specific justification, it is better to use only one kind of internal control. Moreover, a more suitable internal control in this case would be GAPDH (36 kDa), which is sufficiently distant from the PHD2 band.
Response 3: We also appreciate your careful review of the manuscript. Following your recommendation, all instances of ‘kd’ have been corrected to ‘kDa’ in the immunoblotting diagram,fig2a, 4a and 5b. We apologize for the mislabeling.
In the immunoblotting experiments presented in this study, both the target protein and internal control were probed on the same membrane. The membrane was first incubated with the antibody against the target protein, and after use a chemiluminescence imaging system to capture the signal, the antibody was removed using a stripping buffer. The membrane was then re-incubated with the antibody for the internal control, followed by capture the signal and analysis.
Regarding the choice of internal control, we initially used ‘α-tubulin (55 kDa)’ as the control protein. However, we found that its band was too close to the PHD2 target band (50 kDa), which limited its utility. As a result, we considered an alternative internal control. We initially chose ‘GAPDH (36 kDa)’, but found that its expression was influenced by hypoxic stimulation (Regulation of endothelial cell glyceraldehyde-3-phosphate dehydrogenase expression by hypoxia J. JBC, 1994, 269(39): 24446-24453.). Consequently, we switched to ‘β-actin (42 kDa)’, which served as the internal control for the subsequent experiments. We apologize for the lack of rigour in the experimental design due to the use of different internal references in this study, and we are grateful to the reviewers for pointing out this omission and helping us to correct the error. Additionally, the immunoblotting experiment in Figure 4a has been repeated with β-actin as the internal control, and the updated results are shown in manuscript.
Comments 4: In Figures 1b and 5F, if the author is representing the Q2 population, the use of a slash ("/") between the two markers is inappropriate. The y-axis should be labeled as CD71⁺CD235a⁺, not CD71⁺/CD235a⁺, to correctly represent the double-positive population.
Response 4: We would like to express our sincere gratitude to the reviewer for highlighting the inaccuracy, and we sincerely apologize for the error in our writing. As per your suggestion, we have updated the Figure 1b and 5f by changing all instances of ‘CD71⁺/CD235a⁺’ to ‘CD71⁺CD235a⁺’.

Reviewer 2 Report
Comments and Suggestions for Authors
The manuscript "Regulation of PHD2 by HIF-1α in Erythroid Cells: Insights into Erythropoiesis under Hypoxia " reports very interesting findings about the regulation effects by HIF-1α on the PHD2 regulator. The article is well-written and the results are interesting and well described. I have enjoyed this reading and have only very minor comments.-
- Could have been interesting in the introduction to explain that HIF-1α is also quite important in cancer, inflammation and a myriad of other pathologies/lesions (cardiac ischemia) and not only in high-altitude adaptation. Although here the main focus is the erythroid proliferation (thus this focus), but HIF-1α (and its pleiotropic activities) should be properly explain.
- Some sentences in the discussion are kind of repetitive (e.g., lines 237-240, these two sentences could be fused; lines 242 and 243 repeat "does not consistently").
- I really missed a short paragraph stating some/any shortcommings in this research. This is a really nice touch by any author and could be very interesting to know, from the authors' perspective a frank statement about any possible flaw in the design or any other experiment or tests that could have been done to improve the results and further consolidate them.
- I also missed any reference for further applications or next steps/experiments to elucidate more deeply the pathway studied here.
Overall, nice reading and congratulations to the authors.
Author Response
Comments 1: Could have been interesting in the introduction to explain that HIF-1α is also quite important in cancer, inflammation and a myriad of other pathologies/lesions (cardiac ischemia) and not only in high-altitude adaptation. Although here the main focus is the erythroid proliferation (thus this focus), but HIF-1α (and its pleiotropic activities) should be properly explain.
Response 1:
We also appreciate your careful review of the manuscript. Following your recommendation, we have included an introduction to the role of HIF1a in the regulation of cancer, inflammation and a myriad of other pathologies in lines 37-42 in red of the manuscript as follows:
“HIF-1α has been identified as a critical factor in the development and progression of various diseases in numerous studies. HIF-1α contributes to the hypoxic microenvironment in solid tumors[1–3], is implicated in hypoxia-induced cardiovascular diseases[4,5], and plays a role in intermittent hypoxia’s protective effects on the cardiovascular and nervous systems[6], while also being involved in the inflammatory and immune responses associated with bacterial infections[7,8].”
Comments 2: Some sentences in the discussion are kind of repetitive (e.g., lines 237-240, these two sentences could be fused; lines 242 and 243 repeat "does not consistently").
Response 2: Thank you very much for your valuable comments, according to your comments we have made changes in the manuscript, the specific content of the changes are as follows:
“A dual-luciferase reporter assay was designed based on this binding site to confirm whether HIF-1α initiates transcription of the EGLN1 gene in K562 cells by binding to it, and the results confirmed this interaction (Fig. 2e, 2f).” (line269-271)
“Previous research has shown that, although the EGLN1 UTR contains a binding site for HIF-1α, HIF-1α's regulation of EGLN1 transcription is not consistent across cells from different tissues. (line259-261) Additionally, HIF-1α ChIP-seq data showed that HIF-1α does not always bind to the 5' UTR of EGLN1.” (line272-274)
Comments 3: I really missed a short paragraph stating some/any short commings in this research. This is a really nice touch by any author and could be very interesting to know, from the authors' perspective a frank statement about any possible flaw in the design or any other experiment or tests that could have been done to improve the results and further consolidate them.
Response 3: We would like to express our sincere gratitude for the invaluable advice we have received, which has proven to be a profound source of enlightenment. At the inception of this study, it was observed that the protein abundance of PHD2 and HIF1a underwent a change in response to ambient hypoxia. This observation led to the hypothesis that HIF1a might play a regulatory role in PHD2 transcription. Based on this hypothesis, a series of experiments were designed to verify it. However, upon revisiting the paper, it was realized that the effect on PHD2 protein abundance was the sole focus of the transcriptional aspect, while the post-transcriptional and post-translational regulation aspects were neglected.
According to your comments we have summarized the limitations of this study and our plans for future experiments in the final paragraph of the Discussion section, which reads as follows:
"The PHD2 molecule is a key component of the hypoxia signaling pathway, serving as an oxygen concentration receptor. Its protein abundance varies with changes in oxygen levels and the duration of hypoxic exposure, a process likely regulated by multiple factors over time. In this study, we focused on the transcriptional regulation of PHD2 by HIF-1α during acute hypoxic stimulation, but the post-transcriptional and post-translational mechanisms remain unexplored.” (lines 295-300)
Comments 4: I also missed any reference for further applications or next steps/experiments to elucidate more deeply the pathway studied here.
Response 4: We are grateful to you for your perspicacious observations, and we have been inspired by your proposal. According to your comments we have summarized the limitations of this study and our plans for future experiments in the final paragraph of the Discussion section, which reads as follows:
“The reporter model developed in this study, which tracks endogenous PHD2 protein abundance, provides a solid foundation for identifying additional factors that regulate PHD2 levels. In future work, we plan to investigate other potential regulatory factors using established molecular biology techniques, informed by similar studies and complemented by high-throughput sequencing." (lines 300-304)
[1] Fico F, Santamaria-Martínez A. TGFBI modulates tumour hypoxia and promotes breast cancer metastasis J . Molecular Oncology, 2020, 14(12): 3198-3210.
[2] Jing X, Yang F, Shao C, et al. Role of hypoxia in cancer therapy by regulating the tumor microenvironment J . Molecular Cancer, 2019, 18(1): 157.
[3] Bai R, Li Y, Jian L, et al. The hypoxia-driven crosstalk between tumor and tumor-associated macrophages: mechanisms and clinical treatment strategies J . Molecular Cancer, 2022, 21(1): 177.
[4] Richalet J P, Hermand E, Lhuissier F J. Cardiovascular physiology and pathophysiology at high altitude J . Nature Reviews. Cardiology, 2024, 21(2): 75-88.
[5] Semenza G L. Hypoxia-inducible factors: roles in cardiovascular disease progression, prevention, and treatment J . Cardiovascular Research, 2023, 119(2): 371-380.
[6] Guan Y, Gu Y, Shao H, et al. Intermittent hypoxia protects against hypoxic-ischemic brain damage by inducing functional angiogenesis J . Journal of Cerebral Blood Flow and Metabolism: Official Journal of the International Society of Cerebral Blood Flow and Metabolism, 2023, 43(10): 1656-1671.
[7] Bucşan A N, Veatch A, Singh D K, et al. Response to Hypoxia and the Ensuing Dysregulation of Inflammation Impacts Mycobacterium tuberculosis Pathogenicity J . American Journal of Respiratory and Critical Care Medicine, 2022, 206(1): 94-104.
[8] Schaffer K, Taylor C T. The impact of hypoxia on bacterial infection J . The FEBS journal, 2015, 282(12): 2260-2266.
